# Scintillation Characteristics of the Single-Crystalline Film and Composite Film-Crystal Scintillators Based on the Ce^3+^-Doped (Lu,Gd)_3_(Ga,Al)_5_O_12_ Mixed Garnets under Alpha and Beta Particles, and Gamma Ray Excitations

**DOI:** 10.3390/ma15227925

**Published:** 2022-11-09

**Authors:** Jiri A. Mares, Vitalii Gorbenko, Romana Kucerkova, Petr Prusa, Alena Beitlerova, Tetiana Zorenko, Martin Pokorny, Sandra Witkiewicz-Łukaszek, Yurii Syrotych, Carmelo D’Ambrosio, Martin Nikl, Oleg Sidletskiy, Yuriy Zorenko

**Affiliations:** 1Institute of Physics, Academy of Sciences of the Czech Republic, Cukrovarnicka 10, 16253 Prague, Czech Republic; 2Institute of Physics, Kazimierz Wielki University in Bydgoszcz, Powstańców Wielkopolskich Str., 2, 85090 Bydgoszcz, Poland; 3Faculty of Nuclear Sciences and Physical Engineering, Czech Technical University in Prague, Brehova 7, 11519 Prague, Czech Republic; 4Crytur Lt, Na Lukach 2283, 51101 Turnov, Czech Republic; 5CERN, Experimental Physics Department, 1, Esplanade des Particules, 1211 Geneva, Switzerland; 6Institute for Scintillation Materials NAS of Ukraine, 60 Nauky Ave., 61072 Kharkiv, Ukraine

**Keywords:** scintillation, Ce^3+^-doped multicomponent garnets, liquid-phase epitaxy, single-crystalline films, alpha and beta particles, gamma ray

## Abstract

The crystals of (Lu,Gd)_3_(Ga,Al)_5_O_12_ multicomponent garnets with high density ρ and effective atomic number Z_eff_ are characterized by high scintillation efficiency and a light yield value up to 50,000 ph/MeV. During recent years, single-crystalline films and composite film/crystal scintillators were developed on the basis of these multicomponent garnets. These film/crystal composites are potentially applicable for particle identification by pulse shape discrimination due to the fact that α-particles excite only the film response, γ-radiation excites only the substrate response, and β-particles excite both to some extent. Here, we present new results regarding scintillating properties of selected (Lu,Gd)_3_(Ga,Al)_5_O_12_:Ce single-crystalline films under excitation by alpha and beta particles and gamma ray photons. We conclude that some of studied compositions are indeed suitable for testing in the proposed application, most notably Lu_1.5_Gd_1.5_Al_3_Ga_2_O_12_:Ce film on the GAGG:Ce substrate, exhibiting an α-particle-excited light yield of 1790–2720 ph/MeV and significantly different decay curves excited by α- and γ-radiation.

## 1. Introduction

Detectors of ionizing radiation use different scintillators for various applications such as in medical imaging (CT, PET, and PEM [1,2,3,4]), in gamma cameras [5,6], in introscopes at airports for security accessory checks [2], or in X-ray microimaging techniques with micrometer (or even sub-µm) spatial resolution [7,8,9]. Scientific applications include electromagnetic calorimeters of the CMS and ALICE detectors at the LHC in CERN [10,11]. Most of these applications, especially those in medical imaging, should use scintillators characterized by (i) a high light yield (LY) reaching at least 2 × 10^4^ ph/MeV; (ii) fast scintillation decays in the time range of a few tens of nanoseconds; (iii) high effective atomic number Z_eff_ and density above 6 g/cm^3^; (iv) mechanical, chemical, and radiation stability [1,2,5,6,7,8,9,10,11,12].

One of the most investigated and used scintillating crystals fulfilling conditions (i)–(iv) is lutetium aluminum garnet LuAG:Ce (Lu_3_Al_5_O_12_:Ce) [12,13]. The intense development of this crystal started shortly before the year 2000 [14]. At the beginning of this development, Ce^3+^-doped LuAG reached LY values around 10,000 ph/MeV [14,15], but at the end of the year 2010, the LY of this crystal was increased up to 26,000 ph/MeV [16]. LuAG:Ce single crystals were grown from the high-temperature melt using both Bridgeman and Czochralski methods [12,14,17,18]. Due to the fact that the melting point temperature T_melt_ of LuAG:Ce is ~2020 °C [19], the growth of LuAG:Ce crystals at such a high temperature results in the creation of “antisite” defects (ADs) in its structure where the Lu^3+^ cations are embedded at octahedral positions of Al^3+^ cations [20]. The presence of ADs strongly influences the scintillating properties of LuAG:Ce crystals [21], and this is a reason to look for methods of growth in which such ADs will not arise.

Another suitable method of growth of scintillating materials is liquid-phase epitaxy (LPE). Using the LPE method, a large variety of single-crystalline film (SCF) scintillators and multilayered film-crystal composite scintillators were prepared [22,23,24,25,26,27,28]. As SCF garnet scintillators were grown at substantially lower temperatures around 1100 °C, no ADs can arise.

YAG:Ce is another well-known scintillating crystal that was used and studied for more than 50 years [14]. The mixed (Lu_x_Y_3-x_)AG:Ce (x = 1–3) scintillating crystals were prepared as well [17,29]. All these crystals have a cubic garnet structure [2,14,15,18]. Given the flexibility of the garnet structure during the last decade, so-called multicomponent (mixed) (Lu,Gd,Y)_3_(Al,Ga)_5_O_12_:Ce garnets were prepared and intensively studied [30,31,32,33,34,35,36,37,38,39,40,41,42,43,44,45,46,47].

These mixed garnet crystals were grown either by the Czochralski method or by the micro-pulling-down technique [47,48]. Detailed investigation of the scintillating response and properties of multicomponent garnets showed that they can reach very high LY values up to 50,000 ph/MeV, energy resolutions better than 6% at 662 keV, and a reasonable nonproportionality [47]. In addition to bulk crystals, SCFs of various multicomponent garnets were prepared by the LPE method as well [49,50,51,52].

In high-resolution radiography or even microradiography using X- or γ-rays, the SCF thin scintillating screens are very important [7,8,9]. Such radiography can imagine very small objects of the few micrometer or even sub-micrometer dimension range [7]. The current status of LPE technology allows the preparation of both “classical” SCF (one epitaxial layer on the inactive crystal substrate) or composite scintillators (CSs) of Phoswich-type [53]. These scintillators consist of two or even three LPE grown epitaxial layers onto an optically inactive or active substrate. Composite scintillators are grown layer by layer onto the same substrate [54].

To study the scintillating response of the composite scintillator, we can use various radiation sources based on energetic particles or X(γ)-rays [55]. There are differences between the interaction of (i) α- (He^2+^) and β-particles (fast electrons) or (ii) X- or γ-ray quanta, with the scintillation material [55]. The important parameter for the above-mentioned interaction and the following scintillating event (response) is the penetration depth, which is the distance where particles or quanta are fully stopped [55,56,57]. Generally, low penetration depth occurs for α- and β- particles in the range around 10–12 μm (for all garnets) or ~200–400 μm (for LuAG), respectively, while X- or γ-ray quanta generally interact deeper (in the mm-cm range except for low-energy X-rays with energies below 10 keV) [55,57].

In composite scintillators of the Phoswich structure, these differences may be exploited. α-particles interact exclusively in thin surface layers, while γ-photons interact almost exclusively at greater depths, in different layers. If both scintillation materials exhibit significantly different decays, pulse shape discrimination can be used for particle identification.

The main objective of this paper is to present a review of the investigation of the scintillation response and properties of composite scintillators and crystal substrates (Light Yield (LY), energy resolution (ER), and nonproportionality of LY and scintillation decay kinetics) [58,59]. We present the results obtained on GAGG:Ce substrates and composite scintillators containing SCFs of Ce^3+^-doped mixed (Lu,Gd)_3_(Ga,Al)_5_O_12_ garnets. Various types of α- and β-particles and γ-rays were used to generate scintillation (alpha, beta, and gamma spectroscopy) [55,57]). A short summary is presented regarding scintillating properties and characteristics of selected scintillators such as Gd_3_Ga_2.5_Al_2.5_O_12_:Ce substrates (GGAG:Ce), SCFs of similar garnet compounds, prepared from PbO or BaO fluxes [53], and also composite scintillators consisting of three epitaxial layers of YAG:Ce (17 μm), TbAG:Ce (74 μm), and GGAG:Ce (3 μm) garnets, and LPE grown onto the GGAG:Ce substrate. A possible application of these and similar Phoswich structures lies in the aforementioned particle identification by pulse shape discrimination. In this study, it is verified whether the studied samples exhibit scintillation parameters required for the application or not.

## 2. Experimental—Scintillating Properties and Ionizing Sources

Scintillation in crystals is a process arising from their interaction with ionizing radiation, which produces luminescence [53]. Basic scintillation properties of the crystals are the LY, ER, nonproportionality of the LY, and scintillation decay [2,17,55,57,58,59]. Pulse height spectra (PHSs) measurement belongs to basic methods used for the quantification of LY, ER, and nonproportionality properties of the crystals. Our setup consisted of hybrid photomultipliers (HPMTs) DEP PP0470 or PP0475B [58,59], a preamplifier, spectroscopy amplifier ORTEC 672, multichannel analyzer ORTEC 927, and a control computer.

Using a HPMT, we measured the number of registered photoelectrons N_phels_ as a function of the energy of the ionizing radiation [54,56,58]. N_phels_(E) is proportional to the number of emitted scintillation photons N_ph_(E):N_phels_(E) ~ N_ph_(E) × QE × CE_eff_(1)
where E is the energy deposited in the scintillator by ionizing radiation, QE is the averaged quantum efficiency of the HPMT photocathode, and CE_eff_ is the total collection efficiency [58,59]. For the used HPMT, CE_eff_ is ~1 if a Teflon tape reflector (or other efficient reflector) is used; otherwise, it is lower, even by 50%. The measurement setup was calibrated by N_phels_ measurements using the observed single, double, and multi-photoelectron peaks [59] for shaping times 0.5, 1, 2, 3, 6, and 10 µs. From the PHSs, we computed N_phels_(E) at the energy E, then N_ph_(E) using Equation (1). Subsequently, the LY value in ph/MeV [58,59] is obtained as LY = N_ph_/E. The PHSs allow us also to determine the energy resolutions in % for various energy lines. The nonproportionality can be evaluated both from N_phels_(E) or LY(E) values normalizing them to the value at the excitation energy of 662 keV (^137^Cs) [14,17,60,61,62]. The overall accuracy of the setup was ±5%.

Figure 1 presents PHSs of the GGAG:Ce crystal substrate for three kinds of ionizing radiation (α- and β-particles and γ-quanta excitation). We observed a single peak with a heavy low-energy tail for α-particles, a continuously shaped response for β-particles, and a typical total absorption peak for the 661.66 keV γ-ray in the scintillation response.

As noted above, we used three kinds of ionizing radiation (α- and β-particles and γ-quanta) to generate scintillation responses of the crystal substrates, the SCF itself, and composite scintillators (see Table 1 summarizing ionizing radiation sources used in the study). Each of those ionizing radiations interacts differently, which results in a significant impact on the response of the composite scintillator system [55]. α-particles exhibit a low range (d_pen_ as penetration depth) and high linear energy transfer (LET). They are unable to penetrate through the epitaxial film into the substrate. Their track is generally a straight line [55].

For the present study, the α-particles range d_pen_ (see Figure 2) is one of the most important parameters. Table 2 summarizes d_pen_ approximations for important commercial and efficient scintillators including multicomponent garnet crystals Gd_3_Al_x_Ga_5-x_O_12_ (x = 0–5) [56].

The β^−^ particles range (^90^Sr/^90^Y source considered) is significantly longer, around several hundreds of micrometers, and their LET is lower. In addition, their spectrum is continuous from 0 to maximum energy E_β,max_; see Figure 1. At higher energies, they pass through the epitaxial film and deposit most of their energy in the substrate. At lower energies, they can deposit all of their energy in the film. Their track can be described as “curly”.

γ-ray quanta of higher energies interact almost exclusively in the substrate, only scarcely in the film. Secondary electrons produced by γ-photons deposit their energy fully in the substrate. Lower-energy photons (X-rays and γ-rays) may interact in the film and in the substrate as well, depending on the film thickness and composition and photon energy. See Figure 2 for a simplified depiction of interactions of ionizing radiation particles.

Table 1 summarizes the used ionizing sources and some of their properties [63]. α-particles emitted by ^241^Am and ^239^Pu with main peak energies of 5.4857 and 5.1555 MeV, respectively, were used. α-particles emitted by the ^241^Am source covered by thin palladium foil (AP2 α-particle source) were also used. Their mean energy was 4.8 MeV; see Figure 3. Please note that ^241^Am intensively emits γ-photons as well (energy 59.54 keV) [63]. Unlike ^241^Am, ^239^Pu emits only a limited number of X-ray and γ-photons and is, therefore, better suited for scintillation studies using α-particles.

The most common β-particle source is ^90^Sr/^90^Y, which exhibits relatively high energy and practically no X- or γ-ray photons. In addition, the half-life is beneficial for practical application. It was used in this study as well.

Gamma spectroscopy can use various γ-ray emitters in the energy range of 5–2000 keV [14,55]; see Table 1 [63]. The most widely used γ-ray source is ^137^Cs with a γ-ray energy of 661.66 keV.

The half-value layer, i.e., the material thickness at which the γ-radiation intensity is reduced by one half, depends on the linear attenuation coefficient μ. μ depends on the γ-ray energy, material composition, and density. See the linear attenuation coefficient for the GGAG crystal in Figure 4, calculated using XCOM program from NIST [64]. At low X-ray energies up to ~20 keV, the GGAG half-value layer is thin, while above this thickness, it substantially increases.

The Ce^3+^-doped multicomponent (Lu,Gd)_3_(Ga,Al)_5_O_12_ SCF was grown (i) by the Czochralski method or the micro-pulling-down method (bulk crystals) [30,31], or (ii) by the LPE method (thin layers) [48,49,50,51,52,53]. Samples were grown by LPE using PbO or BaO-based fluxes [49]. The thickness of the prepared SCFs can be up to 50 μm and, in addition to a single-layer SCF, composite scintillators were also grown consisting of two or three epitaxial layers [53].

The scintillation decays were measured mainly with the excitation by γ-rays of 661.66 keV (^137^Cs) energy, 5.156 MeV α-particles (^239^Pu), and β^-^ electrons from the ^90^Sr/^90^Y source (continuous energies up to 2.2 MeV). The Hamamatsu PMT R375 connected to the oscilloscope (DSOX6002A, Keysight or TDS3052C, Tektronix) and operated in anode current mode was used for detection of the generated scintillation pulses. In the case of γ-ray excitation, the samples were covered with Teflon tape and placed onto the photocathode window of the used PMT. Radioisotopes were placed 2–5 mm above the samples [30,50,59]. For higher efficiency of light collection, the optical grease Dow-Corning Q2-3067 was used for the optical contact between the sample and photocathode window improvement. We used the same grease also for other scintillation measurements. In the case of excitation with alpha and beta radiation, no Teflon tape was used, in order to avoid energy absorption in the tape. The radiation sources were put directly on the sample surface for α radiation or a bit above the sample surface for β radiation.

All the recorded scintillation pulses were transferred to the PC and the curves with obvious artefacts (i.e., two excitations within the measurement window) were deleted. The remaining curves were then aligned in time and averaged. This resulting curve was eventually fitted using the convolution of the multiexponential function model with the measured instrumental response curve (SpectraSolve software). Table 3 summarizes samples investigated including their composition, fluxes used during LPE growth, and thickness of SCFs.

## 3. Measurement Result

### 3.1. GAGG:Ce Substrate

(Lu,Gd)_3_(Ga,Al)_5_O_12_:Ce multicomponent garnet crystals (LGAG:Ce, GGAG:Ce, and GAGG:Ce according to the content of individual elements) have been intensively investigated in last decade [47]. They are also used as substrates for growth of SCFs using the LPE method [50,53]. In this part, we present the scintillating response and properties of one selected multicomponent garnet crystal GAGG:Ce (Gd_3_Al_2.5_Ga_2.5_O_12_:Ce).

The PHSs of the GAGG:Ce crystal under α-particles (5156 keV of ^239^Pu) and γ-ray quanta (661.66 keV of ^137^Cs) are displayed in Figure 5 and Figure 6, respectively. We measured these spectra under different shaping times in the range of 0.5 to 10 μs. LY and ER values were evaluated from these spectra using Equation (1). QE was calculated using radioluminescence spectra, which were not part of the investigation for the study. The radioluminescence spectra were used solely for QE calculation. Figure 7 presents the dependence of LY and ER values on the amplifier shaping time of the studied GAGG:Ce substrate crystal both for α-particles and γ-ray quanta excitation. Under 661.66 keV γ-ray excitation, LY could reach up to 41,400 ph/MeV, and for 4800 keV, the α-particles energy LY was 5620 ph/MeV, which was about 13% of the value observed under the 661.66 keV γ-ray excitation. Under α-particle and γ-ray excitations, the relative differences between LY values at the shortest and longest shaping time were 23 or 20%, respectively. For more detailed results of LY and ER characteristics (on SCF and composite scintillators), see Table 4 and Section 4. Discussion.

Nonproportionality is generally expressed as the LY value dependence on excitation energy [15,16]. Ideally, LY is independent of energy. Nonproportionality is inconvenient for detector energy calibration and is responsible for ER deterioration as well [65]. Nonproportionality for all the used shaping times of the GAGG:Ce substrate crystal is displayed in Figure 8. In the range of γ-ray energy of 30–1330 keV, nonproportionality was fair within 80 to 100%. Please note that the values for α-particles in Figure 8 are not in fact part of the dependence, as α-particles exhibit a significantly higher LET. The relative LY value for α-particles is given only for the purpose of response comparison.

Very important information can be obtained from the scintillating decay profiles under excitation using α- and β-particles and γ-rays. Figure 9 displays GAGG:Ce scintillating decays under the excitation of α- and β-particles and γ-ray quanta. Their approximations were made by multiexponential fits and an example is shown in Figure 10. Here, we used three exponential fits, where the first component characterizes the rise in scintillation response (τ ≈ 6.6 ns), while the second and third components fit the decay part. Generally, decay curves of one material excited by various particles may significantly differ, but it seems that this is not the case for Figure 9. Detailed results of scintillation profiles evaluation are summarized in Table 4.

### 3.2. SCF and Composite Scintillators

For thorough characterization of all components of SCFs + substrate scintillators, scintillation response excitation by α-particles, β-particles, and γ-photons was used. Due to the different nature of interaction, α-particles deposited their energy in the SCF outermost layer, while higher-energy γ-photons deposited their energy in the substrate. β-particles and lower-energy photons deposited their energy at intermediate depths.

The goal of this paper was an investigation of the scintillating response of SCFs and composite scintillators based on (Lu,Gd)_3_(Ga,Al)_5_O_12_:Ce multicomponent garnets grown by the LPE method using different fluxes (PbO or BaO); see Table 3 [54].

Spectra of all samples under α- and γ excitation were similar, and PHSs of the PL 22-8 Lu_1.5_Gd_1.5_Al_3_Ga_2_O_12_:Ce SCF on the GAGG:Ce substrate displayed in Figure 11 and Figure 12 were, in general, shape-representative for other samples as well. In Figure 11, there is a wide peak with a heavy tail on the low-energy side. The peak was excited by 4.8 MeV α-particles. In addition, a peak produced by the interaction of γ-photons of 59.54 keV energy is observable as well. The response excited by 59.54 keV photons probably originated in the substrate. Pulse height spectra of this SCF under excitation by 661.66 keV photons of ^137^Cs are given in Figure 12. Due to the low attenuation of such γ-rays, most of the scintillation arose in the substrate crystal and the observed LY and ER values were close to those observed at the reference GAGG:Ce crystal.

Figure 13 displays the evaluated values of LY and ER of the PL 22-8 Lu_1.5_Gd_1.5_Al_3_Ga_2_O_12_:Ce SCF on the GAGG:Ce substrate composite in the shaping time range of 0.5–10 μs. Under α-particles, we observed LY values in the range of 1790–2720 ph/MeV and the relative difference between the shortest and the highest shaping time was 51.8%. Using 661.66 keV γ-ray excitation, the LY values were between 29,600 and 35,600 ph/MeV and 20.4% was the relative difference observed between 0.5 and 10 μs shaping times.

Figure 14 presents LY and ER values of the YAG:Ce/TbAG:Ce/GAGG:Ce/GAGG:Ce (Y/T/G/G) substrate composite scintillator measured under different shaping times (0.5–10 μs). Observed LY values were between 446 and 777 ph/MeV for α-particles of 5155 keV in energy and between 17,100 and 25,000 ph/MeV for 661.66 keV of γ-ray quanta. The relative difference between LY values at the shortest and the highest shaping times was 74% or 47% for α-particles and γ-ray quanta, respectively. LY values obtained by α-particles excitation were only 3% of the LY values obtained under γ-photons excitation. This suggests a very low LY of the outermost layer of the particular system, Y/T/G/G. For further details, see Table 4 and Section 4. Discussion.

Scintillation decays under various excitations: α (5.16 MeV), β (^90^Sr/^90^Y), and γ radiation (661.6 keV) are further presented for the PL 22-8 Lu_1.5_Gd_1.5_Al_3_Ga_2_O_12_:Ce SCF/GAGG:Ce substrate and one YAG:Ce SCF/TbAG:Ce SCF/ GAGG:Ce SCF/GAGG:Ce substrate composite scintillator; see Figure 15 and Figure 16, respectively. In both figures, we can observe an evident difference among α-particles-excited decay curves and those excited by β-electrons and γ-ray quanta. On the other hand, β- and γ-excited responses were very similar.

## 4. Discussion

Interpreting the results described in the previous section, significant differences in the nature of interaction of α-particles, β-particles, and γ-photons must be considered. LY and ER are investigated by α-particles and γ-photons. The α-particle-excited response is the response of the outermost layer of the SCFs/substrate composite scintillator. The γ-photon-excited response is the response of the thickest part of the system, i.e., the GAGG:Ce substrate. In addition, the α-particle LET is roughly 1000× higher than the LET of electrons (β-particles or secondary electrons produced by γ-photons).

The same is valid for decay curves measurements, which are measured under β-particles as well. In the β-particles experiments presented here, the pulses of the highest amplitudes are selected for further analysis. Such pulses are less affected by noise than lower pulses. These pulses are produced by the most energetic β-particles, which also exhibit the longest range. Therefore, they deposit their energy mostly in the substrate as γ-photons do.

Then, the following simple observations are easy to explain:(a)The shape of the decay curves measured by β-particle and γ-photon excitation of PL-22-8 and the “Composite scintillator” are very similar; see Figure 15 and Figure 16. The response is excited in the substrate and by particles of almost the same LET. In addition, the 1/e values presented in Table 5 are mostly similar.(b)The LY measured under γ-excitation is very similar among all samples with the notable exception of the “Composite scintillator” (lower LY) and bare GGAG:Ce substrate (higher LY). In addition, LY seems to monotonously decrease with increasing SCF thickness. Therefore, some scintillation photon losses, probably caused by absorption or deteriorated light collection efficiency, takes place in the SCF. These losses are significant in the “Composite scintillator” sample due to the complex nature of the sample (SCF on SCF on SCF on substrate).(c)The energy resolution under γ-photon excitation should be inversely proportional to the number of produced photoelectrons (N_phels_), as it is not necessary to consider the so-called intrinsic energy resolution in this study (the material of the substrate is the same). This assumption is valid within the margins of measurement precision.(d)The highest light under α-excitation is exhibited for the GAGG:Ce substrate, followed by PL 22-8 (Lu_1.5_Gd_1.5_Al_3_Ga_2_O_12_:Ce) and PL 16-4 (Lu_1.5_ Gd_1.5_Al_1.5_Ga_3.5_O_12_:Ce); see Table 4. Similar results of LY/composition dependence were observed in the past as well [31,52]. A Tb-containing SCF exhibits the lowest LY values. Radioluminescence spectra of these SCFs reveal weak Tb^3+^ 4f-4f emission peaks, which is probably a reason for the lower LY compared with those without Tb^3+^ ions emission traces. Finally, the ”Composite scintillator“ sample exhibits a low LY value, significantly lower than the expectation for YAG:Ce. However, it is possible that the complex structure of the sample results in the deterioration of the outermost YAG:Ce SCF layer.(e)The energy resolution measured under α-particle excitation is generally quite poor. A roughly identifiable general trend is there: a lower LY results in a worse ER; see Table 4.(f)The decay curve of PL 22-8 (Lu_1.5_Gd_1.5_Al_3_Ga_2_O_12_:Ce) measured under α-excitation (see Figure 15) first decays much faster than the curve excited by β-particles and γ-photons, and a later, more intense, slower component is presented. The result must be an interplay of different compositions and a much higher LET of α-particles. Unfortunately, without a more detailed characterization by various methods, the origin of the difference cannot be convincingly discussed. For other samples, there is a significant difference as well; see Figure 16 and Table 5. Such results suggest the general feasibility of the application of SCF+substrate systems for particle identification by pulse shape discrimination. The decay curve of the “Composite scintillator” is particularly fast, probably due to the fast decay of YAG:Ce, which does not exhibit as many long scintillation decay components as Lu-containing garnets.(g)The ratio of LY values measured with an amplifier shaping time of 10 μs and shaping time of 0.5 μs indicates the presence of scintillation decay components of decay times of units of μs. For γ-photons and for the bare GAGG:Ce substrate under α-excitation as well, it is 111–119%, which is near the values observed before [67]. The only notable exception is the “Composite scintillator” sample, for which it is 146%. Therefore, it is possible that the substrate quality for this sample might be slightly worse, affecting all the other results (e.g., LY of this substrate).(h)The ratio LY(10 μs)/LY(0.5 μs) varies a lot for all samples from 114% to 380%. Such a variability is to be expected due to the variability in SCF composition and probably quality. The results obtained during this study do not allow for deeper discussion of these differences.

## 5. Summary

We study in detail the scintillation properties of composite scintillators, based on the LPE-grown epitaxial structures, containing single-crystalline films and single-crystal substrate garnets. All of the SCFs are grown on a Gd_3_Al_2.5_Ga_2.5_O_12_:Ce:Ce-substrate. All SCFs are Ce^3+^-doped garnets.

Potentially, an interesting and valuable application of the scintillating detectors based on such Phoswich structures is particle identification by pulse shape discrimination. There are several requirements that must be met: (1) thickness of outermost layer (represented here by single-crystalline film) is slightly greater than the α-particle range, (2) thickness of the innermost layer is great enough for efficient detection of γ and β radiation, (3) decay curve of constituent scintillating layers is significantly different, (4) scintillation response of all constituent scintillating layers exhibits high enough intensity, i.e., high yield is sufficient.

The first and second requirements ensure that α-particles interact solely in the outermost layer, while γ-radiation interact only in the innermost layer and with sufficient probability. The third condition allows efficient pulse shape discrimination. The fourth condition enables efficient detection of the produced pulses by electronics and is important for pulse shape discrimination as well.

Our characterization measurements prove that at least for some composition, these conditions are fulfilled by the SCF+substrate structure. Namely, these are the 16 μm thick Lu_1.5_Gd_1.5_Al_3_Ga_2_O_12_:Ce SCF and 32 μm thick Lu_1.5_Gd_1.5_Al_1.5_Ga_3.5_O_12_:Ce SCF. The thinner sample exhibits a light yield of 1790–2720 ph/MeV for α-particles and 29,600–34,300 ph/MeV for γ-radiation. The thicker sample exhibits a light yield of 1050–1610 ph/MeV for α-particles and 32,900–39,200 ph/MeV for γ-radiation.

On the other hand, the α-particle and γ-radiation responses of Tb-containing SCFs were significantly lower, i.e., 220–250 ph/MeV and 73–280 ph/MeV, respectively. Such a low LY is detrimental for particle detection and discrimination. Therefore, these SCF+substrate structure detectors are unsuitable for the considered application.

The case of the complicated structure of YAG:Ce on TbAG:Ce on GAGG:Ce on GAGG:Ce-substrate is minimal. The LY under α-particle excitation of 450–780 ph/MeV is somewhat detrimental for the purpose, yet still acceptable. However, the more complex structure potentially opens wider application possibilities. Therefore, more complex structures are worth further study.

## Figures and Tables

**Figure 1 materials-15-07925-f001:**
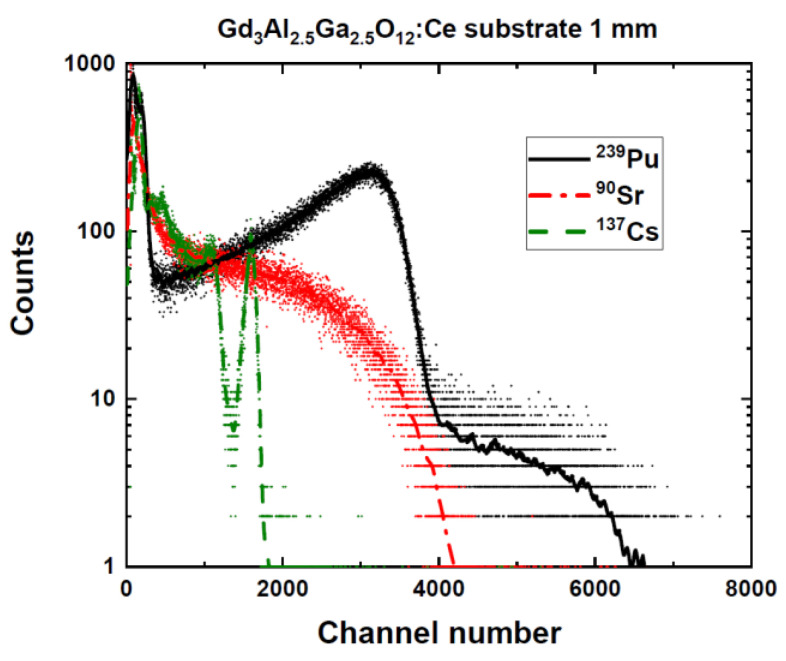
PHSs of GGAG:Ce crystal substrate under excitation of α- and β-particles and γ-ray quanta measured at shaping time of 10 μs.

**Figure 2 materials-15-07925-f002:**
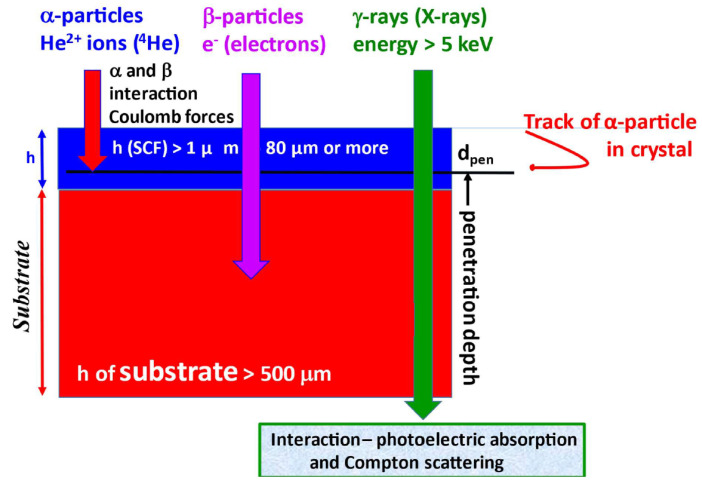
Simplified schema of α- and β-particles and γ-ray quanta interactions in composite scintillator, which stresses the importance of SCF thickness h and α-particle range d_pen_.

**Figure 3 materials-15-07925-f003:**
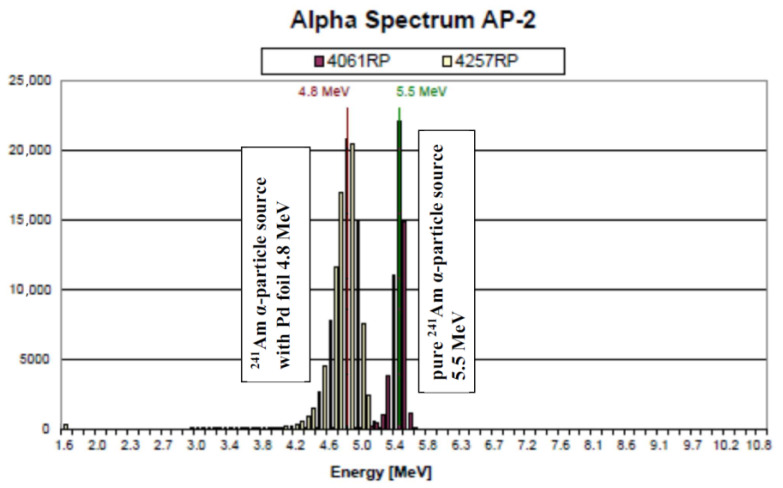
Energy spectrum of two ^241^Am α-particle sources (Amersham, England). Right peak: bare ^241^Am source; left peak: ^241^Am covered by thin palladium foil. This picture is the original one scanned at CERN, Geneva, Switzerland.

**Figure 4 materials-15-07925-f004:**
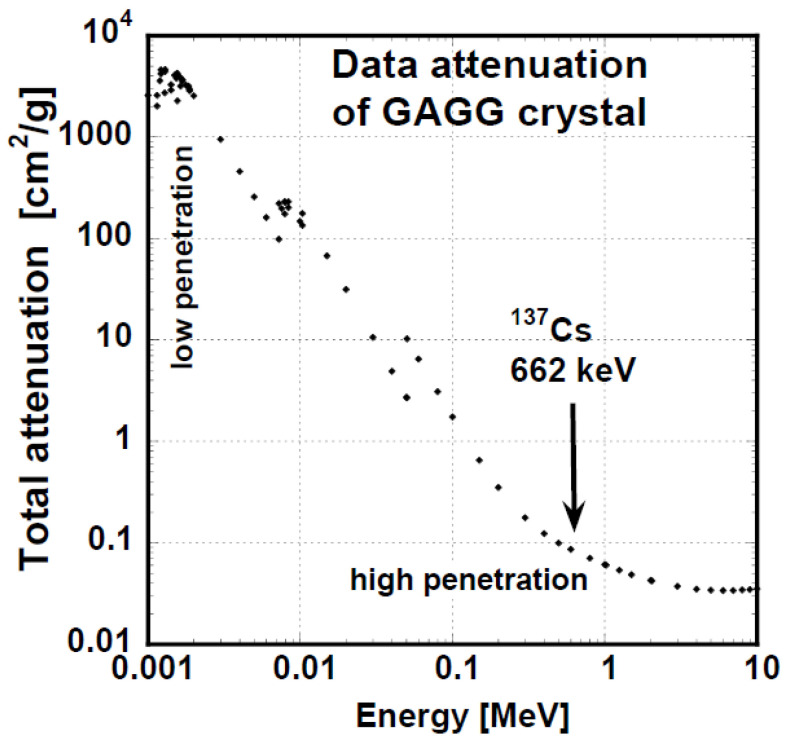
Linear attenuation coefficient μ of GGAG crystal for γ-rays of GGAG crystal (calculated from X-ray and Gamma-Ray Data on www.nist.gov/PhysRefData [64]).

**Figure 5 materials-15-07925-f005:**
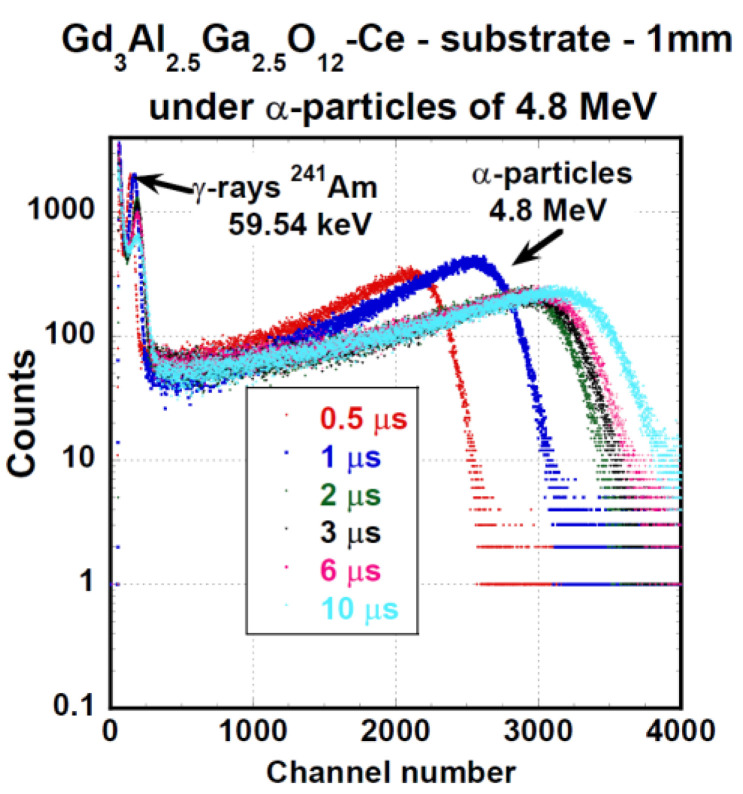
PHSs of GGAG:Ce substrate crystal under α-particles excitation by the energy of 4800 keV (AP-2 source) measured at different shaping time values.

**Figure 6 materials-15-07925-f006:**
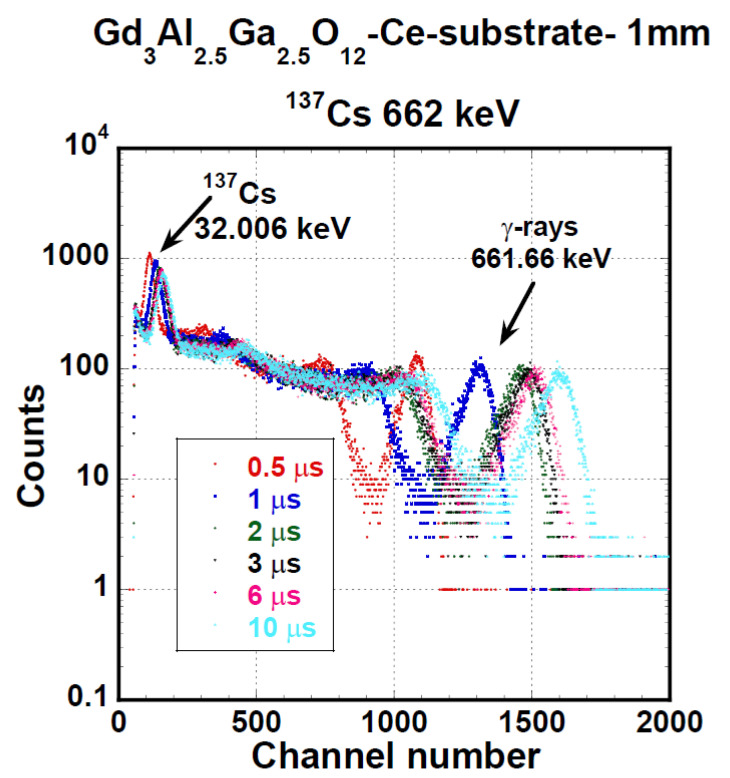
PHSs of GGAG:Ce crystal substrate under γ-ray quanta excitation of 661.66 keV energy of ^137^Cs source measured at different shaping times.

**Figure 7 materials-15-07925-f007:**
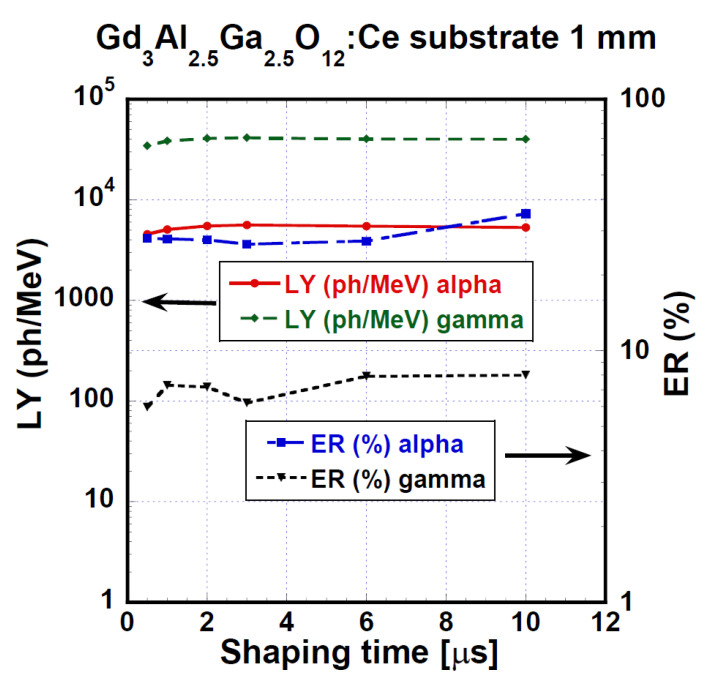
LY and ER of GGAG:Ce substrate crystal under α-particles (4.8 MeV) and γ-ray quanta (661.66 keV) excitations as a function of shaping time.

**Figure 8 materials-15-07925-f008:**
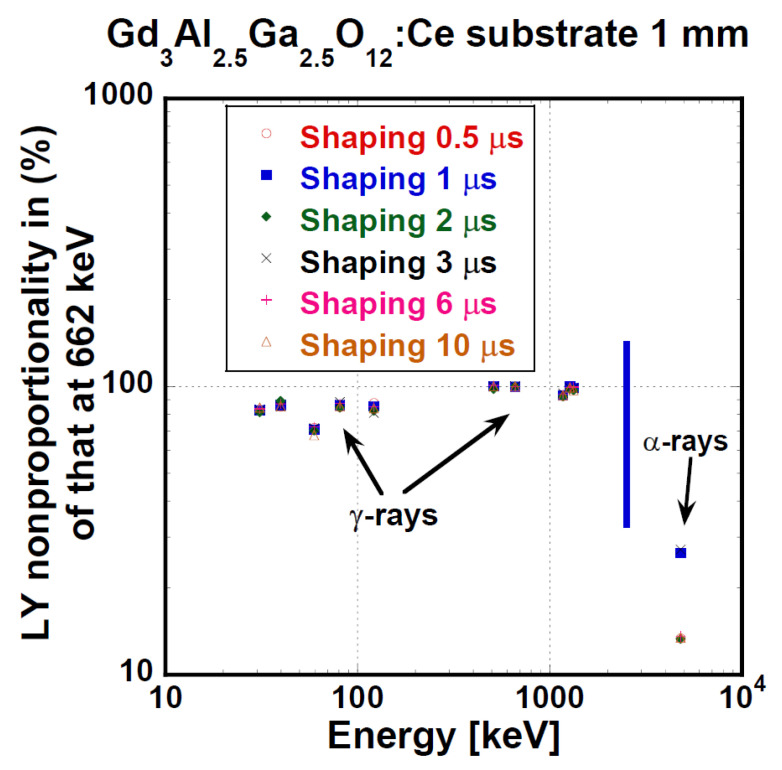
Nonproportionality of LY of GGAG:Ce crystal substrate normalized to that at 661.66 keV as a function of energy and shaping time.

**Figure 9 materials-15-07925-f009:**
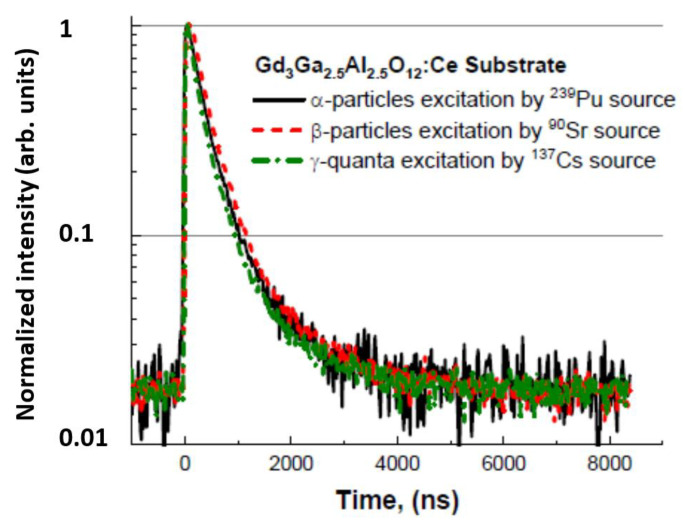
Scintillation decay curves of GGAG:Ce crystal substrate under various excitations by the α- and β-particles and γ-ray quanta (more details see in [66]).

**Figure 10 materials-15-07925-f010:**
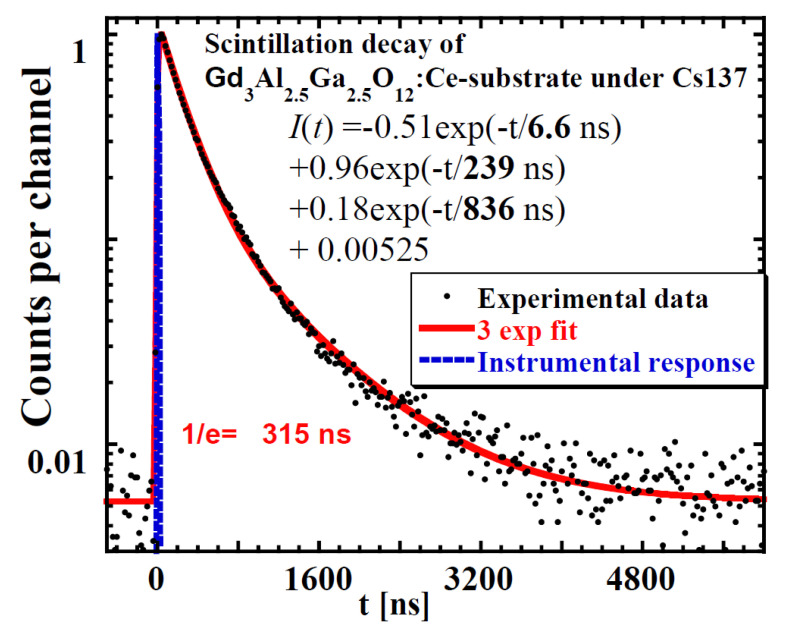
Scintillating decay profile of GAGG:Ce substrate crystal under γ-ray excitation of 661.66 keV energy fitted with 3 exponential components.

**Figure 11 materials-15-07925-f011:**
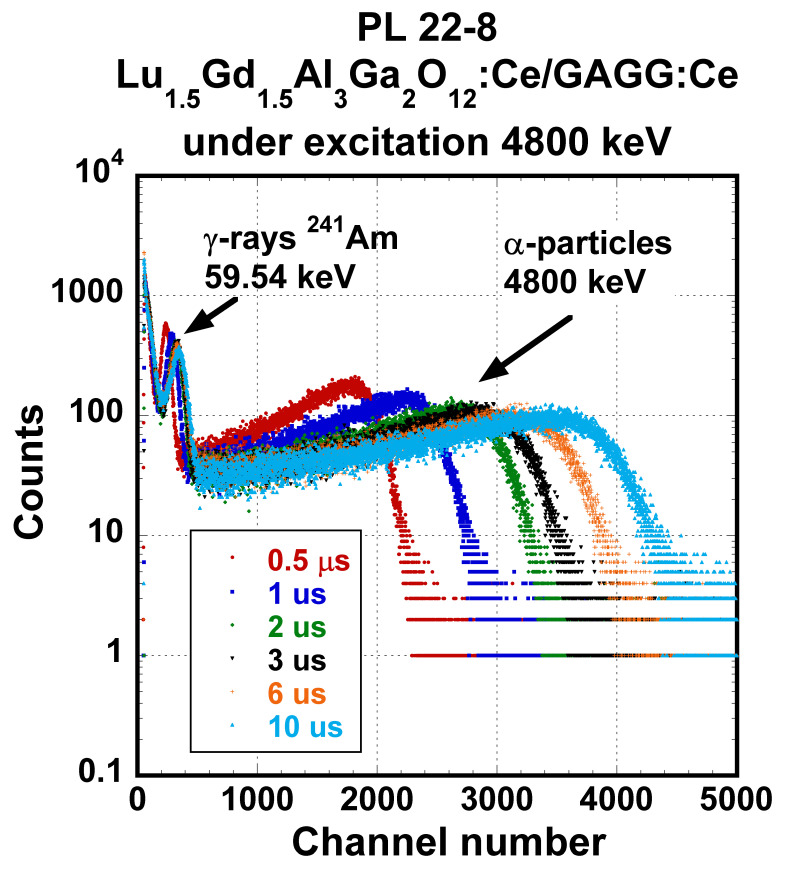
PHSs of PL 22-8 Lu_1.5_Gd_1.5_Al_3_Ga_2_O_12_:Ce SCF /GAGG:Ce SC substrate under α-particles excitation by the energy of 4.8 MeV (AP-2 source) measured at different shaping time values.

**Figure 12 materials-15-07925-f012:**
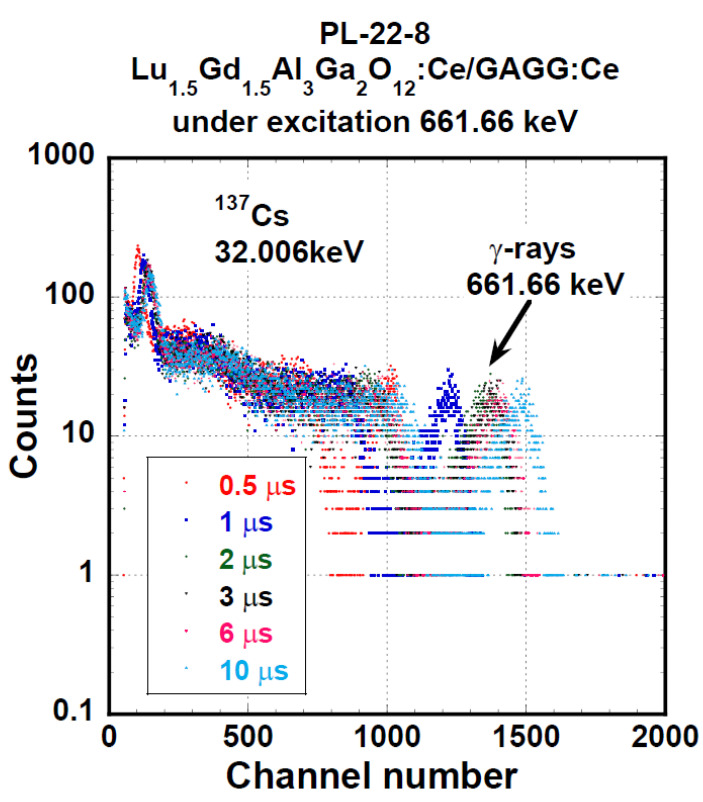
PHSs of PL 22-8 Lu_1.5_Gd_1.5_Al_3_Ga_2_O_12_:Ce SCF /GAGG:Ce substrate under γ-ray quanta excitation with 661.66 keV of energy of ^137^Cs source measured at different shaping time values.

**Figure 13 materials-15-07925-f013:**
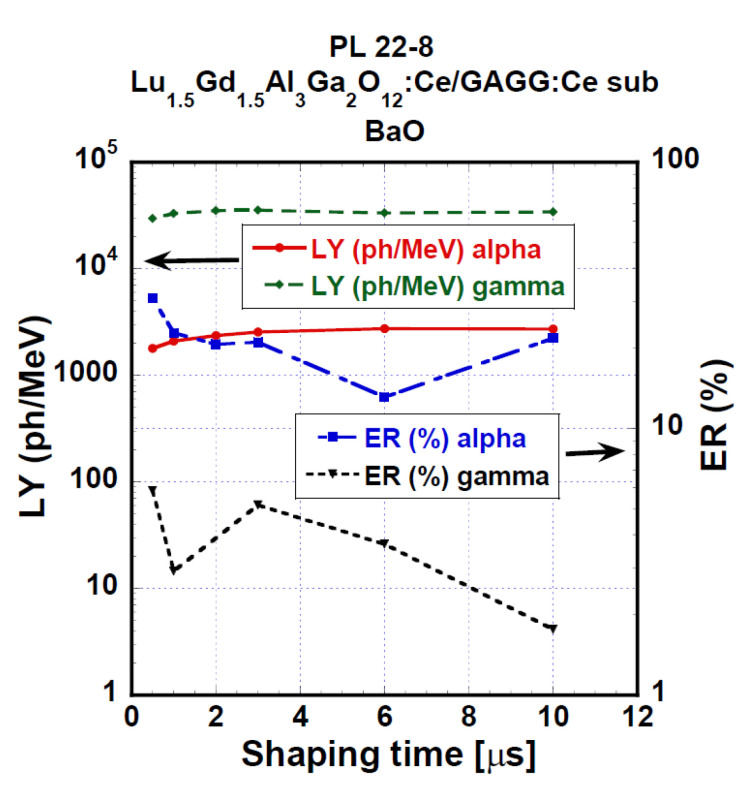
Light yield and energy resolution values as a function of shaping time of the PL 22-8 Lu_1.5_Gd_1.5_Al_3_Ga_2_O_12_:Ce SCF /GAGG:Ce substrate measured in the time range of 0.5–10 μs.

**Figure 14 materials-15-07925-f014:**
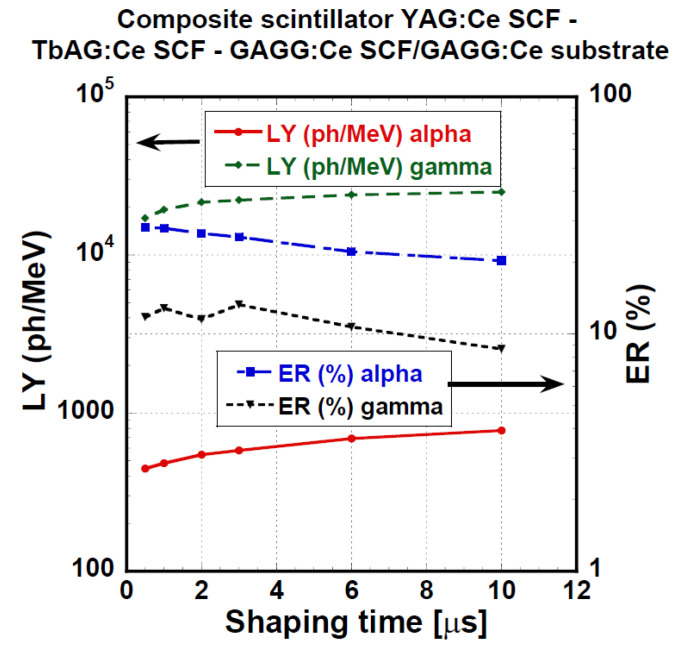
LY and ER as a function of shaping time of the YAG:Ce SCF/ TbAG:Ce SCF/GAGG:Ce SCF/ GAGG:Ce substrate composite scintillator measured in the time range of 0.5–10 μs.

**Figure 15 materials-15-07925-f015:**
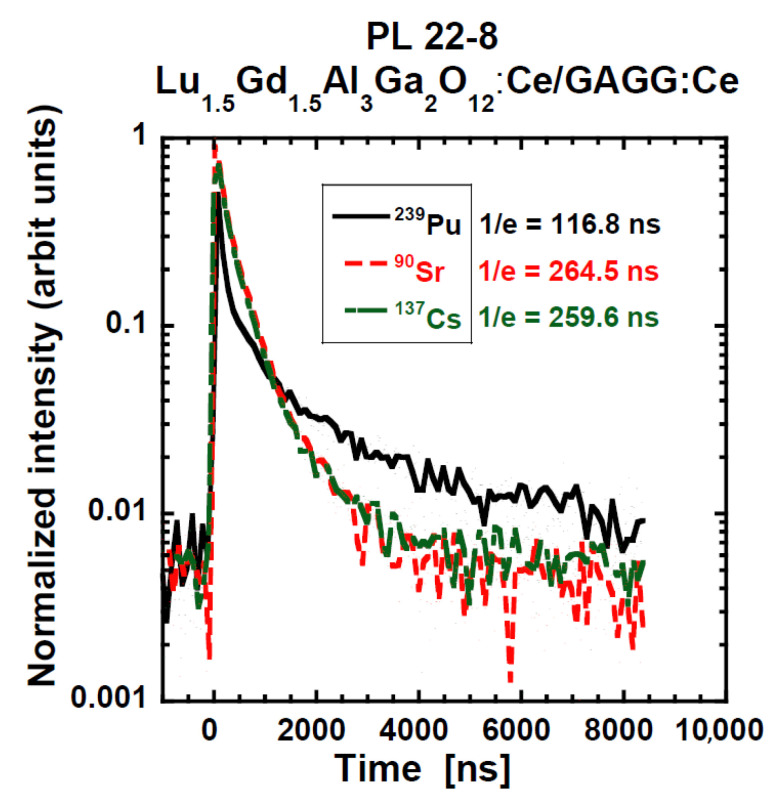
Scintillation decay curves of PL 22-8 Lu_1.5_Gd_1.5_Al_3_Ga_2_O_12_:Ce SCF/GAGG:Ce substrate composite scintillators under various excitations with the α- and β-particles and γ-ray quanta.

**Figure 16 materials-15-07925-f016:**
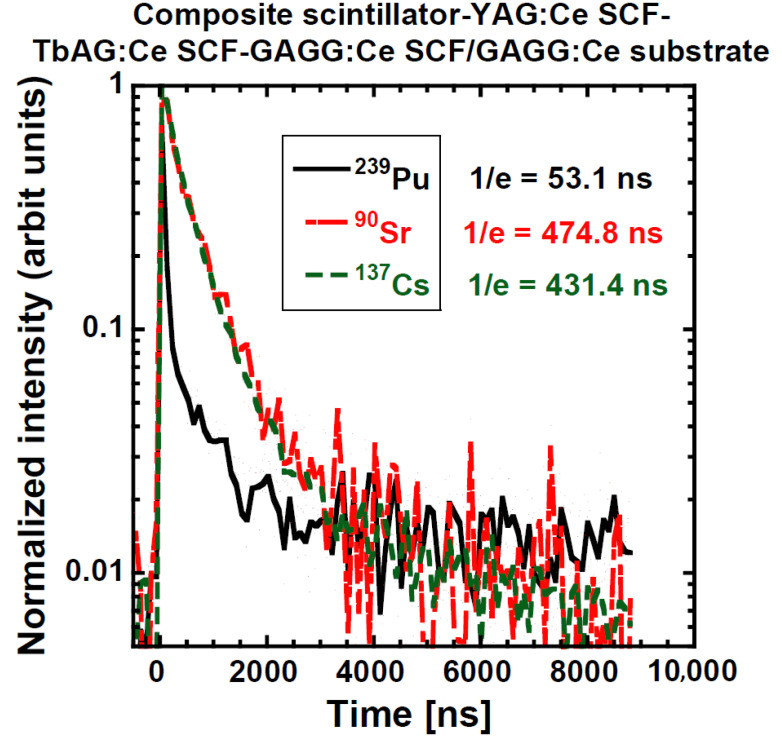
Scintillation decay curves of YAG:Ce SCF/TbAG:Ce SCF/GAGG:Ce SCF/GAGG:Ce substrate composite scintillator under excitations with the α- and β-particles and γ-rays.

**Table 1 materials-15-07925-t001:** Ionizing ionization sources used for measurements of scintillation response of crystals, SCF, and composite scintillators. Y_R_ is radiation yield. Only the most intense energies are presented.

Source	Energy (keV)/Y_R_ (%)	Radiation	Half-Life (years)	Remarks
^241^Am (α)	5442.8/13.0 5485.7/84.5	α-particles	432.5	γ-rays present, mainly 59.54 keV
^239^Pu	5144.3/15.1 5156.6/73.3	α-particles	2.4 × 10^4^	almost no γ-rays present
^241^Am (γ)	59.54/35.9	γ	432.5	the most intense γ-ray energy
^133^Ba	31.0/64.5 81.0/34.1 302.85/18.33 356.02/62.1	X-ray γ γ γ	10.5	the most used X and γ-ray energies
^109^Cd	22.1/55.7 25.0/9.2	X-ray Kα X-ray Kβ	1.3	Low half-life
^57^Co	122.06/85.6 136.47/10.7	γ γ	0.75	Low half-life
^60^Co	1172.2/100 1332.5/100	γ γ	5.27	
^137^Cs	661.66/85.1	γ	30.1	The most used source
^152^Eu	39.91/57.4 121.78/28.6 344.28/26.5	Xray K_α_ γ γ	13.5	the most used energies
Sr^90^/Y^90^	β ^-^_(av)_ 196 (^90^Sr) β ^-^_(max)_ 546 (^90^Sr) β ^-^_(av)_ 939 (^90^Y) β ^-^_(max)_ 2283.9 (^90^Y)	β β β β	28.8 (^90^Sr) 2.7 days (^90^Y)	^90^Y is daughter of ^90^Sr
^241^Am-excited (59.54 keV) characteristic X-ray radiation of different elements *	8.16 13.6 17.8 22.6 32.3 45.5	Cu K_α_ + K_β_ Rb K_α_ + K_β_ Mo K_α_ + K_β_ Ag K_α_ + K_β_ Ba K_α_ + K_β_ Tb K_α_ + K_β_		

***** For details on all radionuclides, see Ref [63].

**Table 2 materials-15-07925-t002:** Penetration depths d_pen_ of α-particles of ^239^Pu (5.155 MeV energy) for selected crystal scintillators. Calculated by approximation formula.

Crystal	LuAG	YAG	Gd_3_Al_x_Ga_5-x_O_12_ (x = 0–5)	GGG	GAG	LSO
d_pen_ (μm)	10.25	11.1	10.4–9.4	9.8	10.7	10.4

**Table 3 materials-15-07925-t003:** Samples, their composition, used fluxes, and thicknesses of SCF used for detailed scintillation measurements.

Sample	Composition	Flux	Thickness (µm)	Remarks
GAGG:Ce crystal	Gd_3_Al_2.5_Ga_2.5_O_12_:Ce	Czochralski method	1 mm	Substrate only
PL 22-8 SCF	Lu_1.5_Gd_1.5_Al_3_Ga_2_O_12_:Ce/ GAGG:Ce-substrate	BaO	16	
PL 25-10 SCF	Tb_1.5_Gd_1.5_Al_2.5_Ga_2.5_O_12_:Ce/ GAGG:Ce-substrate	BaO	63	Weak Tb^3+^ emission lines observed
PL 16-4 SCF	Gd_1.5_Lu_1.5_Al_1.5_Ga_3.5_O_12_:Ce/ GAGG:Ce-substrate	PbO	32	
PL 19-10 SCF	Tb_2_GdAl_1.5_Ga_3.5_O_12_:Ce/ GAGG:Ce-substrate	PbO	33	
Composite scintillators	YAG:Ce(17 µm)/TbAG:Ce (74 µm)/GAGG:Ce(3 µm/ GAGG:Ce-substrate	PbO	YAG SCF 17 µm TbAG:Ce SCF 74 µm GAGG:Ce 3 µm	

**Table 4 materials-15-07925-t004:** Light yields and energy resolutions of GAGG:Ce substrate, selected SCF, and one of the composite scintillators.

Sample	Composition	LY (ph/MeV) α—4800 keV γ—66,166 keV	ER (%) α—4800 keV γ—661.66 keV	Remarks
GAGG:Ce crystal substrate 1 mm	Gd_3_Al_2.5_Ga_2.5_O_12_:Ce	α—4570–5310 γ—34,500–39,600	α—28–35 γ—6–8	γ -rays nonprop.~80% at 30 keV
PL 22-8 SCF	Lu_1.5_Gd_1.5_Al_3_Ga_2_O_12_:Ce/ GAGG:Ce-substrate	α—1790–2720 γ—29,600–34,300	α—13–31 γ—around ≈6	
PL 25-10 SCF	Tb_1.5_Gd_1.5_Al_2.5_Ga_2.5_O_12_:Ce/GAGG:Ce-substrate	α—220–250 γ—28,600–31,800	α—28–35 γ—4–8	Tb^3+^ peaks slightly observed
PL 16-4 SCF	Gd_1.5_Lu_1.5_Al_1.5_Ga_3.5_O_12_:Ce/ GAGG: Ce-substrate	α—1050–1610 γ—32,900–39,200	α—24–39 γ—5–7	
PL 19-10 SCF	Tb_2_GdAl_1.5_Ga_3.5_O_12_:Ce/ GAGG:Ce-substrate	α—73–280 γ—28,700–33,600	α—51–83 γ—6.8–8	
Composite scintillators	YAG:Ce/TbAG:Ce/GAGG:Ce/GAGG:Ce-substrate	α—450–780 γ—17,100–25,000	α—20–28 γ—8.6–13	

**Table 5 materials-15-07925-t005:** Decay properties of GAGG:Ce substrate, selected SCFs, and one of composite scintillators (decompositions into 2 or 3 exponentials were carried out).

Sample (Exact Composition, see Table 3)	Radiation	1st Exponential ns (%)	2nd Exponential ns (%)	3rd Exponential ns (%)	1/e Value
GAGG:Ce substrate	α-4800 keV β-^90^Sr/^90^Y γ-661.66 keV	237 (83.6) 310 (88) 239 (84.2)	816 (16.4) 990 (12) 836 (15.8)	- - -	414 460 211
PL 22-8	α-4800 keV β-^90^Sr/^90^Y γ-661.66 keV	99 (90) 193 (75.3) 203 (81.2)	1640 (10) 688 (24.7) 763 (18.8)	- - -	117 265 260
PL 25-10	α-4800 keV β-^90^Sr/^90^Y γ-661.66 keV	223 (8) 84.3 (34.4) 122 (62.2)	457 (34.3) 268 (55.6) 387 (34.9)	2270 (2) 799 (10.1) 1230 (3)	294 206 190
PL 16-4	α-4800 keV β-^90^Sr/^90^Y γ-661.66 keV	38.3 (36.7) 242 (83.1) 73.3 (34.7)	153 (53.1) 837 (16.9) 295 (56.8)	1650 (10.2) - 967 (8.5)	114 300 211
PL 19-10	α-4800 keV β-^90^Sr/^90^Y γ-661.66 keV	336 (87) 250 (80.4) 134 (70.9)	1300 (13) 825 (19.6) 617 (29.1)	- - -	431 318 207
Composite scintillator	α-4800 keV β-^90^Sr/^90^Y γ-661.66 keV	46 (91.1) 443 (94.6) 387 (90.1)	510 (7.6) 2440 (5.4) 1510 (9.9)	8869 (1.3) - -	53.1 475 431

## Data Availability

We published obtained data in this paper and we published data and their explanation in scientific journals–various our papers and results are in our references e.g. in 50–54, 56,57.

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
