# Peer review of "Scintillation Characteristics of the Single-Crystalline Film and Composite Film-Crystal Scintillators Based on the Ce3+-Doped (Lu,Gd)3(Ga,Al)5O12 Mixed Garnets under Alpha and Beta Particles, and Gamma Ray Excitations"

_materials, 2022, doi:10.3390/ma15227925_

Round 1

Reviewer 1 Report

The authors demonstrated in the details the scintillating properties of composite scintillators under alpha, beta particles and gamma ray excitations, based on the LPE grown epitaxial structures, containing single crystalline films and single crystal substrate of Ce3+-doped multicomponent (Lu,Gd)3(Ga,Al)5O12 garnets. The obtained results are valuable for the exploration and development of scintillators. The manuscript can be accepted for publication in the Materials after some minor revsions.

1.     In the Abstract, some important conclusions should be provided.

2.     Line 340, page 14, “were” should be “are”; Line 187, page 6, “is used” should be “was used”. please check other similar errors in the manuscript.

Reviewer 2 Report

In Table 2, what is the meaning of 241Am and 133Ba listed with half-lives identical to portrayed energies of resp. 16.6 keV and 35 keV? This requires further clarification. In the last entry for 241Am in Table 2, a number of elements are listed (Cu, Rb, Mo, Ag, Ba, Tb); are these meant as decay products of spontaneous fission (SF) of 241Am? If so, a reference is needed.

A thorough clean-up of typesetting is required, for example line 178 where superscript is missing for 239Pu. There are also many spelling mistakes, such as "half-live" instead of "half-life" and "dauter" instead of "daughter" in Table 2, "substrat" instead of "substrate" in figures 5-6, etc.

As a rule of thumb, any figures that are not photographs should be vector graphics that scale properly with resolution. This applies in particular to figures 1 and 3 which are raster graphics with inadequate resolution. Scanning from a low quality figure, as is mentioned in the description of figure 3, is not excusable: in this case, the illustration should be traced in vector software such as Inkscape.

In figures 5-8, how is the shaping time adjusted during measurements?

Reviewer 3 Report

REVIEW

The paper concerns the multicomponents scintillators. In my opinion, the composite film/crystal is very interesting. The authors present new results about the Light Yield, Energy Resolution, nonproportionality and decay kinetics. The topic is very appealing in the framework of new generation scintillating materials. The structure is complex, sometimes a bit confusing, and difficult to read.

1)    As a general observation, in the reasons for the work, I suggest highlighting the advantages of multilayer composites

2)    The samples tested are sometimes confused, to make order in the matter I suggest to put the samples in a table with the structural and compositiona carachteristics, thickness etc. in addition and apart to table 3 at the beginning of section 3.

3)    Fig. 8. Nonoproportionality: what is the unit? (Fig. 8 ) in the text is expressed in % (line224)

4)    Equation 1. Correct the operator of proportionality (not infinity simbol) Nph (E) as Nphels(E) Nph(E) × QE × CEeff,

5)    Line 315 the sentence seems to be interrupted

6)    TABLE 2. The caption is not clear, the second line (179) seems a refuse

7)    Figure 4. Improve the figure. The figure is not sufficiently explanatory
